# Reference Values and Correlations for Multiple Physical Performance Measures: A Cross-Sectional Study among Independently Mobile Older Men in Japan

**DOI:** 10.3390/ijerph17072305

**Published:** 2020-03-30

**Authors:** Yoshinori Ishii, Hideo Noguchi, Junko Sato, Hana Ishii, Ryo Ishii, Shin-ichi Toyabe

**Affiliations:** 1Ishii Orthopaedic & Rehabilitation Clinic, 1089 Shimo-Oshi, Gyoda, Saitama 361-0037, Japan; hid_166super@mac.com (H.N.); jun-sato@hotmail.co.jp (J.S.); 2Kanazawa Medical University, School of Plastic Surgery, 1-1 Daigaku Uchinada, Ishikawa 920-0253, Japan; hanamed12@gmail.com; 3Sado General Hospital, 161 Chikusa Sado, Niigata 952-1209, Japan; kmuyakyu@gmail.com; 4Niigata University Crisis Management Office, Niigata University Hospital, Niigata University Graduate School of Medical and Dental Sciences, 1 Asahimachi Dori Niigata, Niigata 951-8520, Japan; toyabe@med.niigata-u.ac.jp

**Keywords:** older men, healthy life span, physical performance, reference value, walking ability, cognitive function

## Abstract

Purpose: Japan is one of few countries with a male life expectancy over 80 years. The gap between the healthy life span and the total life expectancy is large, highlighting the importance of maintaining physical performance. The present study aims to establish reference values for multiple physical performance measures among high-functioning oldest-old Japanese men and to investigate the correlations among these measurements to understand how these variables are related. Methods: This study was conducted with 120 Japanese males aged 80 years or older who were able to walk independently. Seven measures of physical performance were assessed: handgrip strength, quadriceps strength, static balance ability (one-legged stance), dynamic balance ability (Functional Reach Test; FRT), walking ability (5-m walking time test), combined movement ability (Timed Up & Go test), and bone quality. Cognitive function was also measured (Mini-Mental State Examination; MMSE). Results: Specific reference values are reported for each physical performance measurement explored in this study. Only six participants were classified as cognitively impaired, and 16 had mild cognitive impairment. There were significant correlations of varying levels among all of the measures of physical performance. Age was significantly correlated with all performance measures except FRT, and there was no correlation between age and MMSE. MMSE was weakly correlated with FRT and unrelated to the other performance measures. Conclusions: The reference ranges can be used by older men who have not yet reached 80 years and their health care providers as physical performance targets to facilitate the maintenance of independent mobility in later life.

## 1. Introduction 

According to the Ministry of Health, Labour and Welfare of Japan, the average life expectancy in Japan in 2018 was 81.25 years for men and 87.32 years for women, both of which were record highs [1]. The healthy life span (the period during which there is no restriction on daily life caused by mobility limitations) was 72.14 years for men and 74.79 years for women as of 2016 [1]. This shows that there is a large gap between life expectancy and healthy life span. Several previous studies have reported a link between poor physical performance and mortality. Nofuji et al. [2] found that lower scores on several measures of physical performance, such as walking speed, grip strength, and standing balance, were associated with mortality in a general population of older adults in Japan. In a population-based cohort study in England, Nüesch et al. [3] reported a positive association between the severity of walking disability and the risk of death among patients with knee or hip osteoarthritis. In another cohort study, Imura et al. [4] reported that postoperative ambulatory levels at hospital discharge reliably predicted survival rate among older adults after hip fracture. Considering these previous findings, improving physical performance among older adults may lead to reduced mortality, both individually and at the population level in aging countries such as Japan. Therefore, reference values for measures of physical performance among older adults who are able to walk independently should be clarified for use as target values, both to narrow the gap between the life expectancy and the healthy life span and to extend life expectancy.

However, only a few reports [5,6] have conducted simultaneous evaluations of multiple physical performance measures and tested the correlations among these measures. Previous studies have included one or two physical performance measures, and these were usually conducted among community-dwelling older adult men and women of various ages (i.e., in their 60s, 70s, or 80s and older) and with different levels of walking ability [7,8,9,10,11,12,13,14,15]. Most of these studies have evaluated the correlations between cognitive function [5,7,8,11,12,13] or age [6,9,10,11] and measures of physical performance; few studies [16] have reported the correlations among performance measures for older adults. Measuring multiple types of physical performance simultaneously and examining the correlations among these measures will allow us to evaluate the characteristics of these measures and the links among them in a more detailed and objective way, compared with existing work.

Notably, there are only 11 countries with a male life expectancy above 80 years, although this is more common for female life expectancy [1]. In terms of gender differences in physical performance, a few previous studies have reported different patterns of age-related decline for men and women [6,9,11]. In addition, functional recovery after traumatic hip fracture has been found to be better among men than among women [17]. Conversely, the mortality rate after hip fracture was found to be higher for men than for women [18,19]. 

Therefore, this study aims (1) to establish reference values for multiple physical performance measures that have been reported to be related to the level of ambulation and the risk of falling among men aged 80 years or older who are able to walk independently, and (2) to assess the correlations among seven measures of physical performance, as well as these measures’ correlations with age and cognitive function, which have been examined in previous studies [5,6,7,8,9,10,11,12,13]. In terms of the clinical significance of this work, older adults aged under 80 years may be able to use the reference values established in this study as targets, facilitating the maintenance of independent walking as they age. For this reason, and in contrast to previous studies of community-dwelling older adults, the main objective of the present study is to define reference ranges for physical performance variables among a high-functioning cohort.

## 2. Materials and Methods

### 2.1. Participants

This prospective cross-sectional study was conducted from September 2018 to August 2019. The study included data on 120 Japanese male patients who visited an orthopedic specialty outpatient clinic (median age: 84 years, 25th percentile: 81 years, 75th percentile: 86 years, range: 80–94 years). These patients received diagnoses of degenerative joint and/or cartilage disease. More specifically, the diagnosis was spine-related for 58 patients, upper limb-related for 14 patients, lower limb-related for 43 patients, and related to other areas for 5 patients. To be included in the study, patients had to be men aged 80 years or older who were able to walk independently without support from others, with or without a T-cane. The exclusion criteria were as follows: (1) neurological findings, such as motor paralysis, (2) cognitive or mental dysfunction requiring medication that induced negative effects on physical performance, or (3) skeletal dysfunction that had a negative impact on walking. This study was conducted based on the guidelines laid down in the Helsinki Declaration and the ethical guidelines of our institution. The Research Board of Healthcare Corporation Ashinokai, Gyoda, Saitama, Japan, approved the study (ID number: 2018-5). All patients provided written, informed consent for participation. The patients’ background characteristics are shown in Table 1.

### 2.2. Physical Performance Measures

#### 2.2.1. Muscle Strength

*Upper extremity: handgrip strength.* Handgrip strength (HGS) was measured in the dominant hand using a digital dynamometer (T.K.K.5401 GRIP D, Takei Scientific Instruments Co., Ltd., Niigata, Japan) (minimum measurement unit: 0.1 kgf, accuracy: ±2.0 kgf). Participants were instructed to inhale deeply and fully exhale while squeezing the dynamometer with as much force as possible with their dominant hand. The best result of two trials was used for the analysis. This value was divided by body weight (BW) in kg to eliminate the effect of physique. This ratio of muscle strength to BW (HGS/BW ratio; kgf/kg) was also used in the analysis (GSA).

*Lower extremity: quadriceps strength.* To assess quadriceps strength (QS), isometric knee extension muscle strength in N was measured with the knee in approximately 20° of flexion using a Locomo Scan dynamometer (Alcare Corp., Tokyo, Japan), following a standard protocol [10]. Two measurements of QS were taken, and the highest value on each side was used in the analysis. These values were divided by BW to eliminate the effect of physique. The resulting ratio of QS to BW (QS/BW ratio; N/kg) was used in the analysis (QSA). For the assessment of the correlations among the performance measures, for each participant, the side with the highest QSA was used.

#### 2.2.2. Body Balance

*Static balance ability with eyes open*. One-legged stance (OLS) with eyes open was assessed using the participant’s preferred leg. Participants were asked to place their hands on their waists while staring at a mark on the wall, raise one leg, and stand for as long as possible. They were timed until they lost their balance or reached the maximum time of 60 seconds. Participants performed two trials, and the longer time (to the nearest 0.1 second) was used in the analysis.

*Dynamic balance ability.* The Functional Reach Test (FRT) was used to measure dynamic balance ability. Each participant was instructed to lift his preferred upper limb to a horizontal position and to reach as far as possible without taking a step or touching the wall. During this assessment, the participants’ elbows were extended, their forearms were in neutral pronation/supination, and their hands were open with fingers extended. The initial and maximal reaches were measured in cm from the end of the third finger using a meter stick. The measure was performed twice, and the longer of the two reaches was used in the study.

#### 2.2.3. Movement Ability

*Walking ability.* The 5-m walking time test was used to assess walking ability. This test was conducted as an in-room test consisting of an 11-m linear walking exercise comprising an initial 3-m acceleration zone, a central 5-m timed zone, and a final 3-m deceleration zone. The 5-m walking time started when the patient’s lower limb crossed the starting line between the acceleration zone and the 5-m timed zone and ended when the patient crossed the end line between the 5-m timed zone and the deceleration zone. The measures taken were the time it took the patient to walk the entire 5-m zone at normal and maximum speeds (walking time in seconds), as well as the distance (5 m) divided by the walking time (walking speed; m/second). In this study, both normal walking speed (5-m NWS) and maximum walking speed (5-m MWS) were used in the evaluation of correlations among the performance measures.

*Combined movement ability.* The Timed Up & Go (TUG) test was used to measure combined movement ability. For this test, participants were instructed to stand from a seated position in a chair, walk 3 m, turn, walk back, and sit down in the chair again as quickly as they could. The measure taken from the TUG test was the total time in seconds it took for a participant to complete the test and return to the starting seated position.

### 2.3. Assessment of Bone Property

Bone quality was assessed by bilateral calcaneus broadband ultrasound attenuation (BUA; dB/MHz) using an AOS-100SA (Hitachi-Aloka Medical, Ltd, Tokyo, Japan). All measurements were taken with the same device. Two transducers (receiving and emitting) faced with rubber coupling pads were placed in direct contact on either side of the patient’s heel. Ultrasound gel was applied to the coupling pads to ensure good contact. Standard protocols for this instrument were used to obtain the BUA, including calibration and positioning of the participants. For each participant, the higher value of the two sides was used in the study. The intraclass correlation coefficient for calcaneus BUA, calculated to assess test–retest reliability, was 0.988 (95% confidence interval: 0.955–0.997).

### 2.4. Assessment of Cognitive Function

We evaluated the patients’ level of cognitive functioning quantitatively using the Mini-Mental State Examination (MMSE). This widely used screening test for mental status is designed to evaluate orientation, registration, recall, and attention in older adults. Scores range from 0 to 30 points. The patients were classified as cognitively impaired (<20 points) or not cognitively impaired (≥20 points) [20]. In addition, mild cognitive impairment (MCI) was defined as scoring 23 points or less [21].

### 2.5. Statistical Analysis

Data for certain variables did not pass the Kolmogorov–Smirnov or Shapiro–Wilk test of normality. We used the Wilcoxon signed-rank test to determine whether there was a left–right difference in continuous variables. We used Spearman’s rank correlation test to evaluate correlations between variables. The strength of the correlation of the rank coefficients was defined as strong (0.70–1.0), moderate (0.40–0.69), or weak (0.20–0.39) [22]. In all tests, a *p*-value <0.05 was considered statistically significant. All statistical analyses were performed with IBM SPSS Statistics, Version 23 (IBM Japan, Tokyo, Japan). Values are expressed as medians (25th, 75th percentile; range).

## 3. Results

### 3.1. Descriptive Statistics

The median MMSE score was 26 (25, 29; 16–30) (Table 1). Six participants were regarded as cognitively impaired (scoring <20 points), and 16 participants had MCI.

With regard to muscle strength, we explored both absolute values and weight-adjusted ratios. The median HGS was 29 kgf (25, 33; 12–45), and the median GSA was 0.49 kgf/kg (0.40, 0.55; 0.24–0.69) (Table 1). For the right side, the median QS was 302 N (229, 359; 82–679), and the median QSA was 4.9 N/kg (3.8, 6.1; 1.4–10.9); for the left side, these measures were 312 N (241, 382; 91–669) and 5.3 N/kg (3.9, 6.4; 1.5–0.1), respectively. There was no statistically significant difference between sides (*p* > 0.05; Table 1).

The median static balance ability, as assessed through OLS with eyes open, was 14 seconds (5, 39; 1–60). Dynamic balance ability, measured using the FRT, had a median value of 27 cm (21, 31; 5–48; Table 1).

Five-meter normal and maximum walking ability were measured in relation to both time and speed. The median 5-m NWS was 1.2 m/second (0.9, 1.3; 0.3–1.9), and the test took 4 seconds (4, 6; 3–19). The median 5-m MWS was 1.5 m/second (1.2, 1.8; 0.3–2.8), and the test took 3 seconds (3, 4; 2–6). The median TUG test score, which was included to assess combined movement ability, was 8 seconds (7, 10; 4–23; Table 1).

The median BUA was 57 dB/MHz (47, 64; 23–109) in the right calcaneus and 56 dB/MHz (44, 65; 16–94) in the left calcaneus, and there was no statistically significant difference between the two sides (*p* > 0.05; Table 1).

### 3.2. Correlations among the Measures 

With regard to the associations among the measures of physical performance, there were significant correlations of varying levels for all the measures (Table 2). A strong positive correlation was observed between 5-m MWS and 5-m NWS (*p* < 0.0001, *r* = 0.847), and a strong negative correlation was observed between 5-m MWS and the TUG test (*p* < 0.0001, *r* = -0.776) (Table 2). Both measures of muscle strength (GSA and QSA) showed weak positive correlations with static (OLS) and dynamic (FRT) body balance (GSA vs. OLS: *p* < 0.001, *r* = 0.380; GSA vs. FRT: *p* < 0.0001, *r* = 0.339; QSA vs. OLS: *p* < 0.012, *r* = 0.229; QSA vs. FRT; *p* < 0.0001, *r* = 0.329) (Table 2). The other physical performance measures were moderately correlated with each other (Table 2). TUG test score was negatively correlated with the other measures of performance; the other performance measures were positively correlated with each other (Table 2).

We observed significant positive or negative correlations between age and all performance measures except FRT, and there was no correlation between age and MMSE. The only correlation observed between MMSE and a performance measure was a weak correlation with FRT (Table 2).

## 4. Discussion

The most important contribution of this study was the simultaneous evaluation of seven physical performance measures among Japanese men aged 80 years or older who were able to walk independently, revealing the reference values for these measures in this study population. Following previous studies, we also evaluated the correlations between cognitive function and physical performance [5,7,8,11,12,13] and between age and physical performance [6,9,10,11]. To the best of our knowledge, the present study is the first to examine correlations among multiple physical performance measures, finding various levels of correlation. We found negative correlations between age and all performance measures except FRT, and age was not correlated with MMSE. MMSE was found to be weakly correlated with FRT and unrelated to the other six performance measures.

Cognitive status is considered a predictive factor for healthy life span among older adults—especially in terms of physical activity and walking level [23,24,25]. Matsueda et al. [24] reported a significant relationship between walking level and cognitive status evaluated by the MMSE among patients with hip fracture: Significant differences were observed in mean MMSE by walking status among people in their 80s (dependent walking status: 5.9 points, partially dependent walking status: 16.5 points, independent walking status: 23.7; *p* < 0.0001). In the present study, all participants were able to walk independently, and 95% (114/120) were categorized as not cognitively impaired using the MMSE. Taken together, our findings, along with those from previous reports, indicate that cognitive status can be viewed as an important predictor of maintaining physical activity levels among men aged 80 years or older. However, we did not find significant correlations between MMSE and most of the examined physical performance measurements. This is in contrast to the many studies that have reported significant correlations between cognitive function and aspects of physical performance, such as HGS [12,13], QS [7], and walking speed [5,26]. These conflicting results may be explained by the fact that these previous studies included participants with various degrees of physical performance ability, whereas the present study evaluated only participants who were able to walk independently and therefore had a relatively high level of physical performance. Therefore, most participants in our study could be expected to also have relatively high MMSE scores because of their high level of physical performance. This focus on only participants with high levels of physical performance and cognitive ability may have made it difficult to detect a significant association between MMSE and the physical performance measurements.

With regard to muscle strength, HGS and QS have been reported to decline with aging [6,9,11,27]. Likewise, we found weak negative correlations between aging and both GSA and QSA. Seino et al. [6] reported the mean GS as 27.7 ± 5.9 and the mean GSA as 0.49 ± 0.11 for those aged 80–84 years; for those aged ≥85 years, these researchers reported these values to be 23.2 ± 5.3 and 0.45 ± 0.10, respectively. In addition, a few previous studies [27,28] have evaluated QS using the same arthrometer used in this study. For Japanese men aged 80–89 years (*n* = 86), Narumi et al. [27] reported the median QSA to be 5.9 kgf/kg. The values for both GSA and QSA found that the present study were consistent with these previous findings. We believe that the index values of both HGA and QSA reported in the present study can be used as target values for adults aged under 80 years to maintain independent walking ability as they age.

One-leg balance is an important predictor of injurious falls in older persons [15], and a systematic review has demonstrated that it can also be a predictive tool for frailty among community-dwelling older adult populations [29]. For individuals aged >75 years, Seichi et al. [30] proposed a cutoff of 6 seconds for average OLS time to screen older adults for medical interventions or training programs. Seino et al. [6] reported mean OLS values of 26.0 seconds for those aged 80–84 years and 21.9 seconds for those aged 85 years or older. Taking these previous results into account, the participants in the present study can be considered to have maintained relatively good one-leg balance and to be at low risk of falls.

The FRT is commonly used as an indicator of dynamic balance in older adults. Kamide et al. [31] concluded that older age and female sex were negatively associated with FRT value and that height and two-arm reach were positively associated with FRT value. In this study, because FRT was the only indicator of the seven performance measures that was not correlated with age, it was speculated that FRT was influenced more by other factors, such as physique, height, and two-arm reach, than by age. A recent meta-analysis of data from 20 published studies reported that the weighted mean (standard error) of FRT in these previous studies was 27.2 (0.9) cm, which provides a reasonable standard for interpreting FRT performance among community-dwelling older adults [32]. Considering the findings of these previous studies, the median FRT value of 27 cm found in the present study can be considered to indicate that the study participants maintained a satisfactory dynamic balance.

With regard to walking ability, previous studies [5,6,9,26,33] have focused on walking speed. Recently, Grande et al. [34] summarized the existing evidence concerning the associations of slow gait speed with cognitive decline and dementia, drawing on 39 previous studies. Maximum gait speed may be the best walking-ability marker for cognition among older adults [26]. Seino et al. [6] reported mean values of 1.16 m/second for usual gait and 1.73 m/second for maximum gait among community-dwelling people in Japan aged 80–84 years; for those aged 85 years or older, these values were 1.11 and 1.65 m/second, respectively. Amano et al. [35] identified sex, age, and Kellgren–Lawrence grade [36] as factors influencing walking ability such as 5-m walk test and TUG test outcomes. In the present study, both walking time and walking speed fell within the ranges presented in previous reports [2,5,9,26,33,35].

The TUG test is a commonly used screening tool to assist clinicians in identifying patients at risk of falling. This test is recommended for the routine screening for the risk of falls in the guidelines published by the American Geriatric Society and the British Geriatric Society [37]. However, a systematic review of diagnostic accuracy [38] suggests that a single assessment tool such as the TUG test should not be used to identify community-dwelling older adults at increased risk of falls. Whether there is a significant correlation between the TUG test score and falling remains a controversial topic. However, the TUG test might be helpful in detecting cognitive impairment: The mean TUG test time has previously been reported as 13.2 seconds among people in their 80s without MCI [8]. Taniguchi et al. [14] reported a mean TUG test score of 9.0 ± 4.1 seconds for community-dwelling older adults in their 80s. The average TUG test values found in the present study were higher than these previous findings. This discrepancy may be explained by the fact that the participants in the present study might be at a low risk of falling because few participants with cognitive impairment were included.

BUA is recognized as a parameter associated not only with bone density but also with bone architecture and elasticity [39]. Ultrasonographic measurements of the os calcis have been shown to predict the risk of hip fracture among older women with equal accuracy to dual-energy x-ray absorptiometry of the hip [39]. Ishii et al. [40] reported significant improvement from 48 (34,61) to 56 (34,62) in BUA 5 years after TKA surgery, concluding that this improvement might be explained by increased activity levels after surgery, which might increase mechanical loading of the calcaneus among older adult patients. Yanagimoto et al. [41] used BUA to demonstrate that the amount of walking was positively correlated with bone property in older adults. In addition, Yung et al. [42] reported that weight-bearing exercises had positive effects on calcaneal bone properties, as assessed using quantitative ultrasound (including BUA). Comparison of our BUA values with those of a previous report [40], men aged 80 years or older who are able to walk independently may maintain good bone property because of their activity levels.

The present study reveals significant correlations of varying levels among all measured performance variables. Because walking independently requires people to adjust their center of gravity while maintaining good body balance, all physical performance measures in this study should be effectively linked to accomplishing this task. If even one of the seven types of physical performance was not well-functioning, we believe older adults might not accomplish independent walking. In fact, a previous report [13] found that even grip strength, which is often thought to be unrelated to walking ability, was associated with the level of activities of daily living for people aged 80 years or older. In terms of the different strengths of the correlations among the performance measurements, it seems reasonable that walking speed was strongly correlated with the other variables. It also seems reasonable that participants with a fast 5-m NWS would also have a fast 5-m MWS, leading to a positive correlation. Likewise, a fast 5-m MWS was linked to a short TUG time, resulting in a negative correlation between these measures. However, it is difficult to determine the reasons for the differences between weak and moderate correlations among performance measurements in the present study. These differences should be clarified using a larger number of cases in multicenter studies in the future.

This study has several limitations. First, the participants who took part in the study suffered from some skeletal dysfunction caused by degenerative joint and/or cartilage disease. However, because they did not need the support of others to walk, the impact of their disease status on their physical performance seemed to be negligible or small. Conducting our study in this clinical setting also meant that some equipment that is not usually available in research could be used to evaluate QS and bone property. This allowed us to add two rarely explored physical performance measurements to this study. In addition, both handheld dynamometer and bone densitometer are small, lightweight, and portable and can be useful for quantitative evaluation of muscle strength and bone property to check progress towards target values even at small scale facilities. Second, the total number of participants was lower than the samples used in previous studies of community-dwelling older adults [6,9,16]. However, the number of participants who were men aged 80 or older was comparable to the samples of this population subgroup used to evaluate physical performance in previous studies, where the number of participants in this age group ranged from 9 to 68 [8,14,27,29,30]. Third, we did not consider other factors that may impact the physical performance measures, such as age, sex, and race. Finally, this was a single-center study, which may limit the generalizability of the results. Verification of the validity of our results through research at multiple facilities is expected in the future.

Despite these limitations, this is a valuable report showing reference values for the highest number of physical performance measurements to date (previous work has examined a maximum of six physical performance measurements [5,6]), as well as the correlations among them, for 120 men aged 80 years or older who were able to walk independently in one of the most rapidly aging nations in the world. The reference ranges found in the present study can be used by older adults who have not yet reached the age of 80 years and their health care providers as target values to facilitate the maintenance of independent walking by establishing an efficient exercise regimen.

## 5. Conclusions

The present study evaluates physical performance measurements that are related to walking and falls among men aged 80 years or older who were able to walk independently. The study clarified the reference values of these variables, as well as the correlations among them, in this population. The reference values reported in this study can be used as target values for older adults aged under 80 years for maintaining physical performance as they age. Establishing an efficient exercise regimen to achieve these target values seems to be an urgent issue for older adults who fall short of the reference values for the examined indicators of physical performance.

## Figures and Tables

**Table 1 ijerph-17-02305-t001:** Patients’ characteristics and physical performance measurements among Japanese men aged 80 years or older.

*N* = 120	Median (Percentile)
Age (years)	84 [81, 86] (80–94)
Body weight (kg)	60 [56, 66] (43–84)
Body length (cm)	159 [156, 164] (144–177)
Body mass index (kg/m^2^)	24 [22, 25] (17–33)
Mini-mental state examination	26 [25, 29] (16–30)
Grip strength (kgf)	29 [25, 33] (12–45)
weight-adjusted (kgf/kg)	0.49 [0.40, 0.55] (0.24–0.71)
Quadriceps strength	
Right (N)	302 [229, 359] (82–679)
weight-adjusted (N/kg)	4.9 [3.8, 6.1] (1.4–10.9)
Left (N)	312 [241, 382] (91–669)
weight-adjusted (N/kg)	5.3 [3.9, 6.4] (1.5–10.1)
One-leg standing (min)	14 [5, 39] (1–60)
Functional reach test (cm) N = 119*	27 [21, 31] (5–48)
5m Gait speed (m/sec)	1.2 [0.9, 1.3] (0.3–1.9)
Time (sec)	4 [4, 6] (3–19)
Maximum gait speed (m/sec)	1.5 [1.2, 1.8] (0.3–2.8)
Time (sec)	3 [3, 4] (2–16)
Timed up & go (sec)	8 [7, 10] (4–23)
Bone property (Broadband ultrasound analysis; dB/MHz)	
Right	57 [47, 64] (23–109)
Left	56 [44, 65] (16–94)

* One patient did not complete the Functional Reach Test. Values presented as medians [interquartile range] (range).

**Table 2 ijerph-17-02305-t002:** Spearman’s rank correlation coefficients among all physical performance measurements, age, and Mini-Mental State Examination (MMSE).

	Age	MMSE	GSA	QSA	OLS	FRT	5m-NWS	5m-MWS	TUG	BUA
Age	1.000									
MMSE	−0.081	1.000								
	0.381									
Grip strength adj (kgf/kg)	**−0.283**	0.165	1.000							
(GSA)	**0.002**	0.072								
Quadriceps strength adj (N/kg)	**−0.220**	0.164	**0.457 ^a^**	1.000						
(QSA)	**0.016**	0.076								
One-leg standing (sec)	**−0.391 ^a^**	0.134	**0.380 ^a^**	**0.229**	1.000					
(OLS)		0.147		**0.012**						
Functional Reach Test (cm)	−0.162	**0.205**	**0.339 ^a^**	**0.329 ^a^**	**0.407 ^a^**	1.000				
(FRT)	0.078	**0.025**								
5 m normal walking speed (m/sec)	**−0.382 ^a^**	0.043	**0.495 ^a^**	**0.417 ^a^**	**0.470 ^a^**	**0.393 ^a^**	1.000			
(5m-NWS)		0.644								
5 m Maximum walking speed (m/sec)	**−0.334 ^a^**	0.063	**0.426 ^a^**	**0.479 ^a^**	**0.495 ^a^**	**0.507 ^a^**	**0.847 ^a^**	1.000		
(5m-MWS)		0.494								
Timed Up & Go (sec)	**0.366 ^a^**	−0.198	**−0.414 ^a^**	**−0.407**	**−0.518 ^a^**	**−0.487 ^a^**	**−0.685 ^a^**	**−0.776 ^a^**	1.000	
(TUG)		0.031								
BUA (dB/MHz)	**−0.220**	0.055	**0.256**	**0.206**	**0.262**	**0.220**	**0.246**	**0.300**	**−0.305**	1.000
	**0.016**	0.550	**0.005**	**0.024**	**0.004**	**0.016**	**0.007**	**0.001**	**0.001**	

Adj: adjusted for weight, BUA: broadband ultrasound attenuation. The upper row shows the correlation coefficient, and the bottom shows the *p*-value. a = *p* < 0.001, bold numbers indicate a statistically significant correlation (*p* < 0.05, r ≥ 0.20).

## Data Availability

The datasets used and/or analyzed during the current study are available from the corresponding author on reasonable request.

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
