# Peer review of "Reference Values and Correlations for Multiple Physical Performance Measures: A Cross-Sectional Study among Independently Mobile Older Men in Japan"

_ijerph, 2020, doi:10.3390/ijerph17072305_

Round 1
Reviewer 1 Report
IJERPH 754302
This is a well written article that provides reference values for a range of physical performance measures. Despite the limitations on generalisability, the findings may be helpful to readers in other countries where increasing numbers are aged over 80 years.
Introduction
L 42 ‘between life expectancy and healthy lifestyle’ is fine. The definite article ‘the’ is not needed.
L 53 Suggest replace ‘shorten’ with ‘narrow’
L65-68 I’m not sure of the relevance of this information to the current study.
L73-75 I presume this sentence ‘considering these previous findings..’ is there to begin stating the argument for the conduct of the study. I would omit, as the case is made in the following paragraph.
Methods
The measurement procedures are clearly outlined. Whilst it is OK to give acronyms for measures eg HGS, these should be used consistently throughout the text and in the tables. Muscle strength should be written in full, rather than abbreviated.
The reason for presenting values as medians and percentiles must be made explicit.
Bone quality isn’t a standard measure of ‘physical performance’. Perhaps the rationale from the discussion can appear in the methods section.
Table 1 presents results: it should be moved to that section. Column 1 content requires left justifying
Results
The use of two sub-headings, for the descriptive statistics and the correlations respectively, might be helpful.
It is incorrect to present P values of 0.000. I suggest that P <0.001 is used. I would reconfigure Table 2 using superscripts (eg 0.28a a = P <0.01) to flag the P values. In a sample of 120, correlations to 3 decimal points seems unnecessary.
L 206 Insert ‘significant positive or negative’
Discussion
This section is probably too detailed in its reporting of previous studies.
This discussion serves to indicate that the ‘high functioning’ group’s measurements are not extremes, but fall within previously published ranges. There is scope for further deliberation on how the findings can be used, particularly since the article was submitted for the Health Behavior, Chronic Disease and Health Promotion section of the Journal.
It is interesting that only FRT was correlated with MMSE score. As the authors imply, this is probably due to the sample’s homogeneity with regard to cognitive status. There could be discussion of how clinical/practical use of these reference values as ‘targets’ may need to be tempered by some weighting for cognitive decline.
L 289-295 What is the rationale for including this detailed information? What is the case being made?
L311-312 ‘We believe.. ‘ this wasn’t tested here. Whilst you may be correct, unless you have evidence from elsewhere, I’d omit this sentence.
L319 ‘.. affects..’ replace with ‘was associated with’
L331 ‘Conducting our research in this clinical setting’ – correct, in other settings it may not be feasible to administer multiple measurements of physical performance. Given this, consider some discussion of how the modest correlations here may indicate that a smaller set could be used elsewhere to check progress towards ‘targets’.
L 338/9 You did consider age. What do you mean?
L 347/8 – to facilitate … How? A brief mention of evidence based options would be helpful, particularly since the conclusion states that ‘establishing an efficient exercise regimen’, without any prior explicit discussion that this is one route by which target values could be reached.
Reviewer 2 Report
TÍTULO. Reference values and correlations for multiple physical performance measures: a cross-sectional study among independently mobile older men in Japan
The main objective of this work is to establish the reference values in 7 indicators of physical performance in a sample of men over 80 years old who are able to walk independently and autonomously, and who do not show cognitive impairment.
Additionally, the relationships between these different indicators are studied; between the indicators and cognitive impairment and between the indicators and age.
The three objectives are of empirical and applied interest.
The authors highlight as the main value of the study to study simultaneously the 7 indicators of physical performance since in other studies all these indicators have not been taken into account simultaneously.
I wanted to highlight the main merits of the work that lead me to suggest its publication:
First, focusing on healthy older people is of great interest to scholars of normal aging and the aging expectations that people develop.
Second, it shows an interest in studying the expectations of healthy aging among those over 80.
It is of great interest because it focuses attention on functionality versus lack of functionality or impairment.
Third, it focuses on a group of older people who are under-represented in ageing studies and publications. These studies are crucial, because they contribute to the representativeness of real ageing.
Fourth, identifying physical performance benchmarks for a group of functional over-80s is of interest because it reveals possible predictors of healthy aging in the over-80s that, in addition, depend, in part, on modifiable aspects (behavioral habits or lifestyles).
Fifth, potential impact on reducing negative stereotypes and beliefs about ageing. Focusing attention on the physically well-functioning over-80s allows a rethinking of the concept of normal ageing; it allows the effect of physical ageing to be dissociated from other sources of deterioration (e.g. lifestyle or social ageing).
Sixth. Finally, it uses measurement indicators widely used in other studies, which allows the results to be compared with those of other works.
The selection of background is interesting (although not always applicable)
On the other hand, there is no doubt that the present study has important limitations, the main limitations derive precisely from the sample selection criteria.
These criteria have determined that certain variables included in the study do not have true variability, so it has been difficult to capture their effects on physical performance. The absence of cognitive impairment, and the lack of data about health (physical or perceived) limits the interpretation and discussion of the results. On the other hand, being a cross-sectional study and an incidental sample, it is necessary to be very cautious when interpreting these results.
In any case, the merits set out above lead me to recommend that the work be published.
However, I would point out some minor suggestions:
1. Participants. Please, Justify why only participants with medicated cognitive impairment have been excluded and not the rest.
2, In Table 1. It would include the acronyms of the variables
3. In the description of the force indicators (line 114 and line 121 respectively) note the abbreviations of the adjusted indicators (GSA and QSA respectively)
"2.2.1. Muscle strength 107
Upper extremity: handgrip strength. Handgrip strength (HGS) was measured in the dominant hand 108 using a digital dynamometer (T.K.K.5401 GRIP D, Takei Scientific Instruments Co., Ltd., Niigata, 109 Japan) (minimum measurement unit: 0.1 kgf, accuracy: ± 2.0 kgf). Participants were instructed to 110 inhale deeply and fully exhale while squeezing the dynamometer with as much force as possible with 111 their dominant hand. The best result of two trials was used for the analysis. This value was divided 112 by body weight (BW) in kg to eliminate the effect of physique. This ratio of muscle strength (MSt) to 113 BW (MS/BW ratio; kgf/kg) was also used in the analysis (GSA).
…
Lower extremity: quadriceps strength. To assess quadriceps strength (QS), isometric knee extension 115 MSt in N was measured with the knee in approximately 20° of flexion using a Locomo Scan 116 dynamometer (Alcare Corp., Tokyo, Japan), following a standard protocol [10]. Two measurements 117 of QS were taken, and the highest value on each side was used in the analysis. These values were 118 divided by BW to eliminate the effect of physique. The resulting ratio of MSt to BW (MSt/BW ratio; 119 N/kg) was used in the analysis. For the assessment of the correlations among the performance 120 measures, for each participant, the side with the highest MSt/BW ratio was used (QSA)."
4.I think the following statement should be qualified. (Lines 286-287)
“We believe that our results of relatively high values for MMSE and walking speed, even among men aged 80 years or older may support the previously reported positive correlations between cognitive level and physical performance [27,37].”
The relatively high cognitive functionality is not a result, but a criterion that the authors have adopted in the selection of the sample. The low variability in this variable makes it impossible to interpret its effects on performance. On the other hand, as would be expected, the MMSE does not correlate with the rest of the physical performance variables (with the exception of FRT). As an illustration of this, this study also does not allow us to conclude -which is not done in this work- that the absence of neurological findings, such as motor paralysis; or of skeletal dysfunction that had a negative impact on walking correlates positively with physical performance.
Reviewer 3 Report
First I would thank you for the nice work done. I believe that the aim of the paper is fully reached. I understand that it is a group of specific population. I have only one question: a brief questionnaire like SOC-13 or would have been interesting to know something about the life conducted: job?
